## RESEARCH ARTICLE

# Effects of Square-Stepping Exercise on cognitive function in early geriatric rehabilitation: A randomized controlled explorative study

Katja Fränzel[1,2], Jessica Koschate-Storm[2,3]*, Ellen Freiberger[4], Ryosuke Shigematsu[5], Tania Zieschang[2,3], Svenja Tietgen[1,2]

**1** Department of Geriatrics, General Hospital Bremerhaven Reinkenheide gGmbH, Bremerhaven, Germany, **2** Department of Health Services Research, Faculty VI Medicine and Health Sciences, Carl von Ossietzky Universität Oldenburg, Oldenburg, Germany, **3** Department of Geriatrics, Carl von Ossietzky Universität Oldenburg, Oldenburg, Germany, **4** Institute for Biomedicine of Ageing, Friedrich-Alexander University Erlangen-Nürnberg (FAU), Nürnberg, Germany, **5** School of Health and Sport Science, Chukyo University, Toyota, Aichi, Japan

* Jessica.koschate-storm@uni-oldenburg.de

## Abstract

### Objectives

The aim of the explorative study was to evaluate the effect of Square-Stepping Exercise (SSE) on cognitive function compared to conventional physiotherapy (cPT) in early geriatric rehabilitation. A training effect of SSE on cognitive function particularly on executive functions was expected.

### Methods

This explorative study was conducted in the department of early geriatric rehabilitation in a general hospital. Fifty-eight inpatients (27 female), with a median age of 79.1 (range: 63–90) were randomized to the control group (CG, n = 29) or the intervention group (IG, n = 29). CG received cPT five days per week during their hospital stay. For the IG, SSE replaced cPT for at least six sessions throughout the hospital stay, alternating with cPT. Executive function was assessed via the test battery for attentional performance (TAP), memory function was evaluated using the digit span test.

### Results

Both groups improved in the divided attention task (total pre 9.9 missing items; total post 7.5 missing items, $p = 0.011$), and in the flexibility task (total pre 2034.49 ms; total post 1680.60 ms, $p = 0.004$). There was no specific training effect of SSE on executive functions measured with the TAP. No improvement in memory function was noted in either group.

**Data availability statement:** The study was conducted in a hospital. Data was partly taken from patient records. This data cannot be made publicly accessible. Data are available from the Klinikum Bremerhaven-Reinkenheide (contact via svenja.tietgen@klinikum-bremerhaven.de) or uac.medizin@uol.de for researchers who meet the criteria for access to confidential data.

**Funding:** The author(s) received no specific funding for this work.

**Competing interests:** The authors have declared that no competing interests exist.

**Abbreviations:** CG, Control Group; cPT, Conventional Physiotherapy; EQ-5D, EuroQol- 5 Dimension; FES-I, Falls Efficacy Scale – international; HRQL, Health Related Quality of Life; IG, Intervention Group; ITT, Intention-to-treat; MMSE, Mini Mental Status Examination; SPPB, Short Physical Performance Battery; SSE, Square-Stepping Exercise; TAP, Test Battery for Attentional Performance.

## Conclusion

Inpatients receiving SSE combined with cPT improved similarly in cognitive domains during early geriatric rehabilitation compared with inpatients offered cPT alone. SSE can be used as an additional component in early geriatric rehabilitation, which due to its playful characteristics might be more motivational or intriguing for some inpatients.

## Trial registration

ClinicalTrials.gov DRKS00026191

## Introduction

According to the World Health Organization (WHO), healthy aging is defined as the process of developing and maintaining functional abilities with the goal of individual well-being [1]. Functional ability in this context includes physical and cognitive performance, and their interaction which are key determinants of mobility [1]. Impairments in these areas might affect walking ability with an increased risk of falls, which might pose a risk to independence in activities of daily living [2–3]. Therefore, it is imperative to equally train cognitive and physical functioning to maintain mobility throughout the aging process.

Maintaining functional ability after an acute health event is a crucial aspect of rehabilitation in the healthcare system. In geriatric rehabilitation a multidimensional diagnostic and therapeutic approach is key for the treatment of multimorbid inpatients. It is frequently offered after acute in-hospital care in rehabilitation centers. Additionally, in the German healthcare system, early geriatric rehabilitation is established and reimbursed as a specialized treatment for geriatric inpatients encompassing acute care with rehabilitative therapy in a multiprofessional team [4]. The structured program utilizes a multi-professional approach that includes activating nursing care, physiotherapy, occupational therapy, speech therapy, and neuropsychological therapy, with the main objective of maintaining and recovering functional independence, integrating the individual goals of the patient. The length of treatment is two to three weeks determined by the German health care system. The choice of the content of therapy is left to the therapists and is not standardized, which leads to personalized training content. Accordingly, systematic reviews of early rehabilitation in hospitalized older people have shown heterogeneous results concerning the content of interventions and outcomes [5–7]. In addition to differences in intervention content, studies often provide limited information about the duration, frequency, and intensity of the selected methods, further complicating the comparison of interventions [6,8]. Therefore, it is important to investigate standardized programs in the specific setting of early geriatric rehabilitation. These programs should include physical and cognitive aspects in order to maintain mobility in the aging process.

Physical training and simultaneous physical and cognitive training have been shown to have positive effects on cognitive performance in older age in community dwelling older persons [9–11]. There is evidence indicating that simultaneous training

may be more effective than physical training alone or alternating physical and cognitive training [12–14]. Systematic reviews and meta-analyses on this topic have been limited by inconsistent study populations with varying control conditions and non-standardized training programs [13–14]. One training option to integrate physical and cognitive training simultaneously is Square-Stepping Exercise (SSE), a standardized training program. SSE combines motor and cognitive demands. It was originally developed by Shigematsu and colleagues [15] as a group training for seniors in Japan, living independently. The aim was to improve functional fitness of the lower extremities to improve walking ability and reduce the risk of falls. SSE training has been shown to be effective in improving physical function and preventing falls by improving balance in community dwelling older people [15–19]. Meanwhile, the implementation of SSE has also been studied in younger adults [20] and in populations with specific diseases such as Parkinson's disease [21–23], multiple sclerosis [24] and diabetes mellitus [25]. Recently, a pilot study was published by our research group that investigated SSE for the first time in the setting of early geriatric rehabilitation in Germany [26]. In SSE, step patterns are performed in multiple directions on a mat divided into squares. SSE follows the principle of proactive and reactive response improvement and thus practices, for example, a corrective step to circumvent obstacles. The level of difficulty increases as the training progresses. After successful performance of the step patterns, the training can be enhanced with a cognitive component (e.g., word counting, mathematical tasks). Due to the high cognitive demand of the training, a positive effect on cognitive function is expected. A small number of studies on SSE training have focused on cognitive function. A 16-week SSE training program showed an effect on executive function in community-dwelling older adults compared to a control group without intervention [27]. A meta-analysis by Wang and colleagues [28] summarized the effects of SSE on motor and cognitive function and fall risk in their analysis. The included studies varied in the design of the intervention conditions. The authors concluded that the effects of SSE on cognitive function require further research. A comparison between an online SSE condition and a face-to-face condition showed short-term effects of SSE on executive function and group cohesion in sedentary older adults [29]. Consistent with the findings of Teixeira and colleagues [27], a home-based online study by Kawabata and colleagues [20] in sedentary young adults (mean age 22.8, SD = 1.2) showed that SSE improved executive functions such as abstract reasoning, mental flexibility, and problem-solving skills. In a recent study with a limited number of cases, executive functions were enhanced following seven weeks of SSE group training. Additionally, to the best of the authors' knowledge, there are no studies to date investigating the short-term effects of SSE on cognitive functions. However, the evidence base for the effects of SSE on cognitive function is still limited. To our knowledge our trial [26] was the first study to explore the implementation of SSE in the specific setting of early geriatric rehabilitation. Apart from the vulnerable study population a substantial difference is that the two- to three-week training period in this setting is shorter than in previous studies. Nevertheless, we could show, that SSE in alteration with physiotherapy was as effective as physiotherapy only in this specific setting concerning the improvement of functional performance. This paper evaluates the effect of SSE compared with the conventional early geriatric rehabilitation program based on daily physiotherapy on cognitive function. Additionally, concerns about falling (CaF) and health-related quality of life (HRQoL) were assessed as secondary outcomes. The reduction of CaF as well as beneficial effects on HRQL should be considered a valuable indicator of the effectiveness of geriatric rehabilitation as it reflects the core goals of improving self-care, mobility, social interactions and communication [30–31]. Importantly, there is a direct influence of concerns about falling on self-reported HRQL [32]. Due to the high cognitive demands of the SSE and the data available from previous studies, we expected a positive effect on cognitive function.

## Methods

### Study design and setting

This study was designed as a single-blinded, randomized controlled intervention trial. It was conducted in a general hospital in the department of early geriatric rehabilitation in Bremerhaven, Germany in accordance with the Declaration of Helsinki, including its amendments until 2013. This study was approved by the Ethics Committee of the Bremen Medical

Association (number 785). It was registered at the German Clinical Trials Registry (DRKS00026191). All procedures were carried out with written informed consent of the participants.

According to the sample size calculation done for the primary endpoint Short Physical Performance Battery (SPPB) we recruited 58 inpatients in geriatric acute care undergoing early rehabilitation from November 2021 to August 2022 [26]. The following analysis focusses on the secondary endpoints of the study.

Patients, participating in inpatient early geriatric rehabilitation, able to walk short distances (10m) in company, without walking aid, Mini Mental State Examination (MMSE) score >22 to ensure ability to consent and sufficient cognitive skills to implement the training program, and sufficient knowledge of German or English were included in the study. Inpatients with severe aphasia, severe visual limitations, high grade presbyacusis, severe impairment of physical functionality, and limitations of functions of the arms and legs with the inability to walk were excluded. Via block randomization with permuted blocks of variable length participants were randomized to the IG or the CG. A third party not involved in the study assigned the identification number. The assessors were blinded to group allocation. Fig 1 shows the process of screening, recruitment, allocation, follow-up, and data analysis. The original figure and a detailed description of the process have been published previously [26].

The CG received cPT five times a week for 30 minutes each, which corresponds to the conventional program in early geriatric rehabilitation.

Participants in the IG received alternating SSE and cPT. SSE training was performed three times per week and cPT twice per week, each lasting 30 minutes. The ratio of SSE and CP was maintained over the average three weeks of hospitalization. Based on the step patterns and severity classifications of the original studies, published by Shigematsu and colleagues, SSE training was performed according to a standardized protocol [15–16]. Within the study group, a selection was made from the original patterns. At the beginning of each training session, the participants were familiarized with the SSE mat. The training progressed in difficulty, depending on the individual abilities of the participants. Throughout the training sessions, step patterns became more complex. Upon confident completion of the respective step pattern, a cognitive component was added to the stepping exercise. Participants were asked to simultaneously solve a calculation task or enumerate terms. In addition, arm movements (e.g., waving or clapping) were used at predefined points on the mat. Fig 2 shows examples of step patterns for each difficulty level.

## Data collection

Data were collected at two time points. Baseline measurement was performed prior to randomization and post intervention measurement was performed prior to hospital discharge. Investigators were blinded to assigned groups.

## Clinical characteristics

Clinical characteristics, including demographic data and functional status (Barthel Index) [33], were documented from the patient's medical record. Additional data such as years of education, walking aids, and average walking distance were obtained through a standardized interview.

## Assessment of cognitive function

In order to assess possible changes in individual domains of executive function, alertness, divided attention, and flexibility tests from the Zimmermann and Fimm attentional performance test battery (TAP) [34] were used. This is a computerized standardized test battery that is routinely used in the geriatric department. Age-, gender- and education-adjusted standardization is available.

To evaluate *alertness*, simple reaction processing speed is measured. A total of 80 target stimuli is presented at random intervals over a period of 270 seconds. Participants are instructed to respond to these stimuli as quickly as possible by pressing a button. Each participant performs the test in two different conditions twice, with the presentation of 20 target

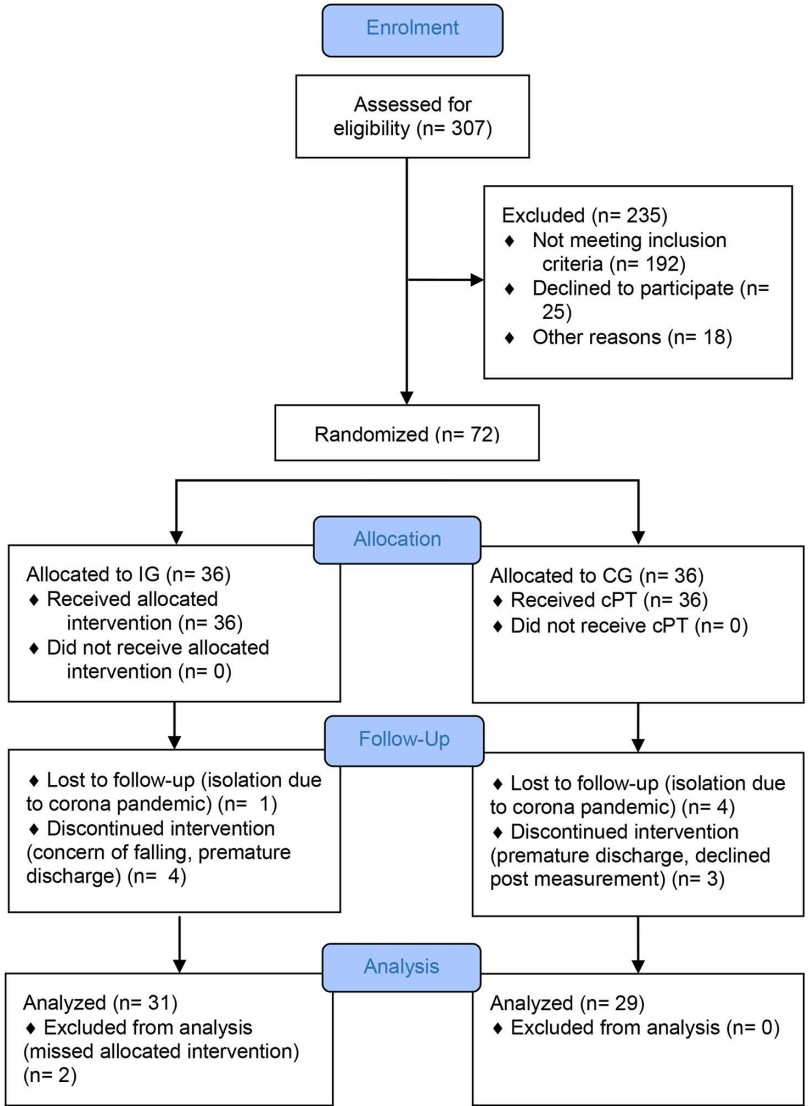

**Fig 1. Participant flowchart [26].** Participant flowchart Abbreviations: IG, intervention group; CG, control group; cPT, conventional physiotherapy.

stimuli each: In one condition, the target stimulus appears on the screen at randomly varying intervals. In the other condition, a warning tone serves as a cue stimulus before the target stimulus appears. Phasic alertness is calculated from the difference between the mean reaction times of the two conditions in milliseconds (ms).

During the test to assess *divided attention* of the participants, a visual and auditory task has to be completed simultaneously: for 17 of the 100 presented visual stimuli and for 16 of the 200 presented auditory stimuli a response by pressing a button is required.

This subtest *flexibility* is a set-shifting task. The participant is asked to respond to changing target stimuli by pressing a button. From trial to trial, the participant must respond to the complementary target stimulus. There are 100 target stimuli presented in a 180-s session. The index value represents the relationship between error rate and reaction time in ms. An index score of +/- 0 indicates average performance. A negative index value indicates below average performance with an increased error rate and/or increased reaction time.

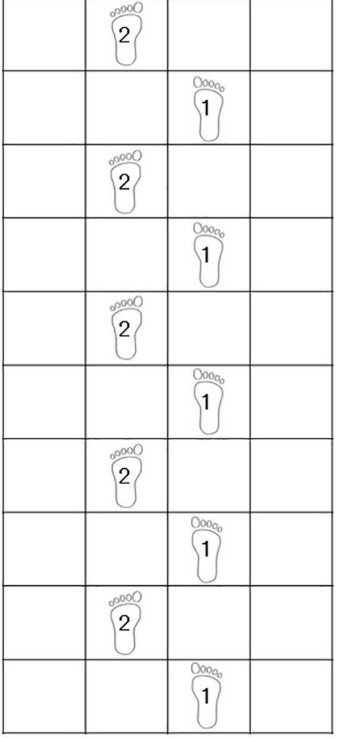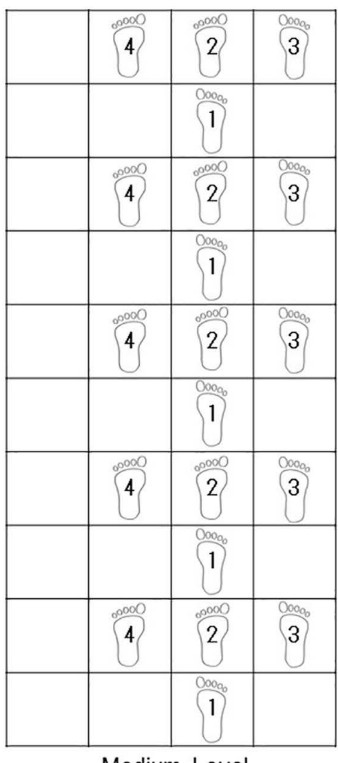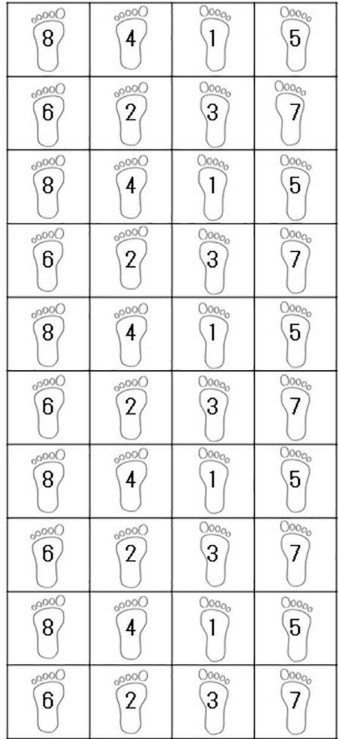

**Fig 2. Examples of step patterns in beginner, medium and advanced level.**

Verbal memory performance was measured using the Digit Span subtest from the Wechsler Adult Intelligence Scale-4th Edition (WAIS-IV). Short-term verbal memory was evaluated by the digit span forward, working memory by the digit span backward test [35].

## Concerns about falling assessment

Concerns about falling was assessed using the 16-item German version of the Falls Efficacy Scale-International (FES-I) [36]. This questionnaire includes concerns about falling during light and heavy physical activities and during social activities. Concerns about falling in each activity is rated on a 4-point scale (1 = not at all concerned, 4 = very concerned). Therefore, higher scores indicate a higher level of concern about falling.

## Assessment of health-related quality of life

The EuroQol-5Dimension (EQ-5D) [37] measures five dimensions: mobility, self-care, usual activity, pain/discomfort, anxiety/depression. The questionnaire also includes a visual analog scale (VAS) that allows participants to rate their perceived health status on a scale from 0 (worst possible health status) to 100 (best possible health status).

## Statistical analysis

Unpaired t-tests and Chi-square-tests were used to compare demographic and clinical characteristics of CG and IG at baseline. A two-factor (time*group) repeated measures ANOVA was used to compare test performance between the two groups for the two time points (baseline = t1 and post intervention = t2). The Greenhouse-Geisser adjustment was used to

correct for violations of sphericity. Post-hoc Bonferroni tests were used if significant main effects were observed. A significance level of $\alpha = 0.05$ was used for all calculations, except for testing for normal distribution and sphericity ($\alpha = 0.1$). All analyses were performed with SPSS version 29.0 (SPSS, Inc., Chicago, IL, USA).

## Results

During the study period, 307 inpatients were screened for eligibility on admission to the department of acute geriatric care. Seventy-two eligible inpatients were randomized to the CG (n = 36) and the IG (n = 36). After randomization seven participants in the CG dropped out (four because of isolation due to SARS-CoV-2 infection, three due to premature discharge at own request and refusal of final examination). In the IG seven participants were excluded after randomization (one because of isolation due to SARS-CoV-2 infection, two due to less than six SSE sessions, one withdrawal of consent, one premature discharge at own request, one refusal of final examination and one concerns about falling). The statistical analysis included a total of 29 participants of the CG and a total of 29 participants of the IG (Fig 1).

The CG received a total number of 12 ± 2 physiotherapy units, the IG 12. ± 2 including 7 ± 1 SSE units ($p = 0.492$). In both groups no fall events occurred during therapy [26].

### Baseline characteristics

No differences were found between the CG and IG at baseline for demographic or results of the geriatric assessment (Table 1). Further results of the geriatric assessment have been previously published [26].

**Table 1. Demographic and assessment results of inpatients at baseline.**

| | CG (n = 29) | IG (n = 29) | p-value |
|---|---|---|---|
| Age [years] | | | |
| mean ± SD | 78.9 ± 6.70 | 79.4 ± 7.10 | 0.791 |
| (min-max) | (65-88) | (63-90) | |
| Women [number] | 13 | 14 | 0.792 |
| Education [years] | | | |
| mean ± SD | 11.5 ± 1.97 | 12.2 ± 3.10 | 0.316 |
| (min-max) | (7-16) | (7-20) | |
| Mini mental status examination [score] | | | |
| mean ± SD | 27.9 ± 1.63 | 27.5 ± 1.95 | 0.386 |
| (min-max) | (23-30) | (23-30) | |
| Barthel Index [score] | | | |
| mean ± SD | 61.9 ± 12.06 | 64.5 ± 13.32 | 0.441 |
| (min-max) | (40-85) | (40-90) | |
| FES-I [score] | | | |
| mean ± SD | 23.7 ± 6.98 | 23.3 ± 7.54 | 0.857 |
| (min-max) | (16-41) | (16-52) | |
| EQ-5D Level of health [%] | | | |
| mean ± SD | 62.5 ± 20.82 | 60.2 ± 21.19 | 0.686 |
| (min-max) | (20-100) | (20-90) | |

Abbreviations: SD, standard deviation; CG, control group; IG, intervention group; FESI-I, Falls Efficacy Scale-International; EQ-5D, EuroQol-5 Dimensions.

## Effects of the intervention on cognitive function

There was no significant time*group interaction in any of the measured variables (see Table 2). Both groups showed a significant decrease in missing items of the divided attention task after the intervention with no statistically significant group difference. Comparison of performance on the flexibility task showed a significant decrease in reaction time and significant improvement in flexibility index in both groups after intervention with no significant time*group interaction.

Table 2. Effects of TAP subtests in CG and IG.

| Cognitive function variables | Baseline | | Post intervention | | Repeated measure ANOVA | | | | |
|---|---|---|---|---|---|---|---|---|---|
| | CG<br>n=29 | IG<br>n=29 | CG<br>n=29 | IG<br>n=29 | Effect | df | F | p | η² |
| Alertness [ms] | 433.52±201.84 | 395.48±109.63 | 423.83±180.21 | 383.97±102.46 | Time | 1,56 | 1.106 | 0.298 | 0.019 |
| | | | | | Group | 1,56 | 0.979 | 0.327 | 0.017 |
| | | | | | Time*group | 1,56 | 0.008 | 0.928 | < 0.001 |
| Alertness warning [ms] | 378.83±98.47 | 369.90±105.84 | 383.93±150.89 | 357.76±84.06 | Time | 1,56 | 0.071 | 0.791 | 0.001 |
| | | | | | Group | 1,56 | 0.440 | 0.510 | 0.008 |
| | | | | | Time*group | 1,56 | 0.426 | 0.517 | 0.008 |
| Phasic alertness [%] | 0.09±0.23 | 0.07±0.15 | 0.09±0.17 | 0.05±0.14 | Time | 1,56 | 0.058 | 0.810 | 0.001 |
| | | | | | Group | 1,56 | 0.811 | 0.372 | 0.014 |
| | | | | | Time*group | 1,56 | 0.067 | 0.796 | 0.001 |
| Divided attention missing [number] | 8.97±5.93 | 10.90±6.98 | 6.66±5.49 | 8.28±6.93 | Time | 1,56 | 6.999 | 0.011 | 0.111 |
| | | | | | Group | 1,56 | 1.640 | 0.206 | 0.028 |
| | | | | | Time*group | 1,56 | 0.028 | 0.868 | <0.001 |
| Flexibility [ms]<br>IG: n=28* | 1985.69 ±1249.91 | 2085.04±987.60 | 1649.52±854.75 | 1712.79±739.15 | Time | 1,55 | 9.218 | 0.004 | 0.144 |
| | | | | | Group | 1,55 | 0.123 | 0.727 | 0.002 |
| | | | | | Time*group | 1,55 | 0.024 | 0.878 | < 0.001 |
| Flexibility fault [number]<br>IG: n=28* | 10.48±9.51 | 9.93±9.31 | 8.45±9.93 | 8.25±9.86 | Time | 1,55 | 2.702 | 0.106 | 0.047 |
| | | | | | Group | 1,55 | 0.027 | 0.870 | < 0.001 |
| | | | | | Time*group | 1,55 | 0.025 | 0.875 | < 0.001 |
| Flexibility index [score]<br>IG: n=28* | −12.10±14.08 | −13.91±11.98 | −4.24±17.04 | −6.62 ± 16.58 | Time | 1,54 | 26.451 | < 0.001 | 0.329 |
| | | | | | Group | 1,54 | 0.311 | 0.579 | 0.006 |
| | | | | | Time*group | 1,54 | 0.037 | 0.848 | 0.001 |
| Digit span forward [score] | 7.76±1.83 | 7.90±1.63 | 7.93±1.85 | 7.76±1.68 | Time | 1,56 | 0.011 | 0.919 | < 0.001 |
| | | | | | Group | 1,56 | 0.002 | 0.968 | < 0.001 |
| | | | | | Time*group | 1,56 | 0.854 | 0.359 | 0.015 |
| Digit span backward [score] | 5.93±2.00 | 6.45±1.90 | 6.24±1.35 | 6.72±1.56 | Time | 1,56 | 1.681 | 0.200 | 0.029 |
| | | | | | Group | 1,56 | 1.63 | 0.207 | 0.028 |
| | | | | | Time*group | 1,56 | 0.006 | 0.939 | < 0.001 |

Abbreviations: CG, control group; IG, intervention group.

*missing value due to termination of the test by the participant.

## Effects of the intervention on concerns about falling and health related quality of life

There was no significant time*group interaction in any of the measured variables (see Table 3). Both groups showed a significant increase in self-reported HRQL after the intervention with no statistically significant group difference.

## Discussion

The aim of the presented explorative study was to evaluate the effect of SSE in combination with cPT on cognition compared to cPT alone in early geriatric rehabilitation. Due to the high cognitive demand, a specific training effect of SSE on executive functions and memory was expected.

Over the course of the rehabilitation treatment, CG and IG improved equally in the variable divided attention and flexibility. Thus, there was no specific effect of SSE on executive functions. However, SSE combined with cPT was not inferior to cPT alone. This conclusion is supported by the significant improvement in the Barthel Index in both groups. The significant increase in HRQL in both groups also supports the assumption of a positive effect of conventional rehabilitation treatment on mobility without an additional effect of SSE.

In spite of the cognitive tasks applied in SSE and having to memorize the patterns, there was no significant difference in change in memory function between the groups. The effects of cognitive stimulation through SSE on attention and memory function assumed by Teixeira and colleagues [27] could not be replicated. When interpreting the results, the different conditions in the intervention group must be considered. In contrast to prior research, the intervention group in this study received a combination of SSE and cPT training, rather than SSE alone [20,27]. The weekly frequency of SSE was identical, with a total of three sessions per week. However, the training duration administered in this study was only 30 minutes, which is substantially lower compared to previous research. Another limitation is the short duration of the intervention due to the specific setting. Teixeira et al. [27] conducted their intervention over a period of 16 weeks. Although in a young population, Kawabata et al. [20] showed short-term effects in cognition, such as mental flexibility, problem-solving skills and abstract reasoning after only 2 weeks of intervention. As the number of studies on the effect of SSE on cognitive function in older people is still limited, further research in this specific research field with a larger study sample and a longer intervention period is required [29]. A recent systematic review by Siqueira and colleagues highlighted the heterogeneous study landscape on the effectiveness of SSE and the resulting gaps in research [38].

The results of the FES-I showed no reduction in concerns about falling. Previous studies have evaluated concerns about falling in context of SSE with heterogeneous results, but the participants in the studies were significantly younger on average than in the present study. In addition, the participants were community-dwelling, therefore no comparability is given [17,39,40]. To the authors' knowledge, no published study has yet shown that geriatric rehabilitation led to a reduction in concerns about falling as measured by the FES-I [41–42].

Table 3. Effects on concerns about falling and health-related quality of life in CG and IG.

| Further variables | Baseline | | Post intervention | | Repeated measure ANOVA | | | | |
|---|---|---|---|---|---|---|---|---|---|
| | CG n = 29 | IG n = 29 | CG n = 29 | IG n = 29 | Effect | df | F | p | η² |
| FES-I [score] | 23.66 ± 6.98 | 23.31 ± 7.54 | 23.59 ± 7.98 | 24.28 ± 8.49 | Time | 1,56 | 0.235 | 0.63 | 0.004 |
| | | | | | Group | 1,56 | 0.009 | 0.925 | < 0.001 |
| | | | | | Time*group | 1,56 | 0.312 | 0.578 | 0.006 |
| EQ-5D [%] | 62.48 ± 20.82 | 60.24 ± 21.19 | 73.03 ± 18.87 | 64.52 ± 16.62 | Time | 1,56 | 9.856 | 0.003 | 0.15 |
| | | | | | Group | 1,56 | 1.409 | 0.24 | 0.025 |
| | | | | | Time*group | 1,56 | 1.766 | 0.189 | 0.031 |

Abbreviations: CG, control group; IG, intervention group; FES-I, Falls Efficacy Scale- International; EQ-5D, EuroQol-5 Dimensions.

It is important to consider the limitations of the study. The study's sample size was calculated based on the primary outcome of physical function, and was not powered for changes in cognitive function. To provide more specific classification of the results, an a priori power analysis, based on the available data, should be performed for further research.

With regard to the study population, it should be noted that SSE is only feasible for inpatients in early geriatric care with relatively high physical and cognitive functionality. A selection is already present due to the inclusion and exclusion criteria. However, the SSE also requires a certain level of cognitive ability.

In general, it is questionable whether a change in cognitive function can be expected after a three-week intervention. The results of a mini review indicate that a significant change in cognitive performance is observed in older adults after at least 12 weeks or at least 1000 minutes of simultaneous cognitive and physical training at a training frequency of 1–3 times per week. Here, heterogeneity in the type, duration, and implementation of the training programs is also mentioned as a limitation [13]. However, in the specific setting of early geriatric rehabilitation, the duration of intervention is limited to three weeks. Furthermore, we question whether the chosen TAP measurement tool is sensitive to this change. The TAP proved to be appropriate in terms of objectivity of administration and selective monitoring of individual components of executive function. The age-, education- and gender-specific standardization is a further advantage over frequently used screening instruments. Because the study took place in a clinical routine, no further assessments were used.

During the measurement, the motivation of the participants to complete the computer-based assessment appeared to be low, as observed by the test supervisor. This may be due to the lack of everyday relevance of the tasks. Another reason could be that the stimulus material was not sufficiently engaging to enhance the motivation of the participants. Additionally, the duration of the cognitive test (approximately 45 minutes) may have been excessively lengthy, potentially leading to fatigue among the participants.

SSE is a stepping exercise to challenge lower limb function. The cognitive task consists of memorizing the pattern and placing the feet accordingly. However, executive functions were assessed with the hands by pressing a button. In further research, this aspect should be considered by selecting additional test instruments that involve the lower limb. Because SSE involves demands on visual spatial perception, this domain should also be considered in further research when measuring cognitive function (e.g., Choice Stepping Reaction Time Test).

## Conclusions

In conclusion, the development of cognitive function during early geriatric rehabilitation did not differ when a part of the cPT sessions were replaced by SSE training. Due to the limitations of our exploratory study further research is needed to confirm this finding. SSE is feasible and not inferior to cPT in geriatric early rehabilitation and may be used as a playful and attractive alternative for some inpatients. Concerns about falling did not increase. However, an improvement in HRQL could be demonstrated in both groups after geriatric rehabilitation.

## Supporting information

**S1 Appendix. Study protocol English translation.**
(DOCX)

**S2 Appendix. Study protocol original language.**
(DOCX)

**S3 Appendix. CONSORT-Checklist.**
(DOCX)

## Acknowledgments

The authors would like to thank the participants for their cooperation, and they also greatly appreciate the assistance of the staff members who were involved in this study. We would like to express our special thanks to the head of the department, Dr. H. Ackermann, for the opportunity to conduct the study in the Department of Geriatrics.

## Author contributions

**Conceptualization:** Katja Fränzel, Jessica Koschate-Storm, Ellen Freiberger, Tania Zieschang, Svenja Tietgen.

**Data curation:** Katja Fränzel, Svenja Tietgen.

**Formal analysis:** Katja Fränzel, Jessica Koschate-Storm, Svenja Tietgen.

**Supervision:** Jessica Koschate-Storm, Ellen Freiberger, Ryosuke Shigematsu, Tania Zieschang, Svenja Tietgen.

**Writing – original draft:** Katja Fränzel.

**Writing – review & editing:** Katja Fränzel, Jessica Koschate-Storm, Ellen Freiberger, Ryosuke Shigematsu, Tania Zieschang, Svenja Tietgen.

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
