## [Decision Letter · Decision Letter 0]

2 Jul 2025

Dear Dr. Koschate-Storm,

Thank you for submitting your manuscript to PLOS ONE. After careful consideration, we feel that it has merit but does not fully meet PLOS ONE’s publication criteria as it currently stands. Therefore, we invite you to submit a revised version of the manuscript that addresses the points raised during the review process.

We look forward to receiving your revised manuscript.

Kind regards,

Mario Ulises Pérez-Zepeda, M.D., Ph.D.

Academic Editor

PLOS ONE

2. For studies involving third-party data, we encourage authors to share any data specific to their analyses that they can legally distribute. PLOS recognizes, however, that authors may be using third-party data they do not have the rights to share. When third-party data cannot be publicly shared, authors must provide all information necessary for interested researchers to apply to gain access to the data. (https://journals.plos.org/plosone/s/data-availability#loc-acceptable-data-access-restrictions)

5. We note that the original protocol that you have uploaded as a Supporting Information file contains an institutional logo. As this logo is likely copyrighted, we ask that you please remove it from this file and upload an updated version upon resubmission.

Reviewers' comments:

Reviewer's Responses to Questions

**Comments to the Author**

1. Is the manuscript technically sound, and do the data support the conclusions?

Reviewer #1: Yes

Reviewer #2: Partly

2. Has the statistical analysis been performed appropriately and rigorously?

Reviewer #1: I Don't Know

Reviewer #2: Yes

3. Have the authors made all data underlying the findings in their manuscript fully available?

Reviewer #1: No

Reviewer #2: No

4. Is the manuscript presented in an intelligible fashion and written in standard English?

Reviewer #1: Yes

Reviewer #2: Yes

Reviewer #1: Overall an interesting manuscript. More major concerns involved the data being collected in 2021-2022 but now being published in 2025 - an explanation for the delay and decision to publish now may be helpful. The 40 references are also all from 2023 or before and the majority were published pre 2020 and therefore pre COVID. It would be helpful to update some of the references to make the manuscript as relevant to the present as possible.

The text describes how. 72 patients were included from 307 patients and Figure 1 indicates that 192 were excluded due to not meeting inclusion criteria. However, it would be helpful to understand which criteria the patients failed if this data is available. If nearly 2/3 of patients were excluded for not meeting the inclusion criteria then greater detail is required to ensure that there was not a selection bias hidden within these exclusions. It would also be helpful if the authors provided some examples of the 18 'other reasons' why patients were not included for randomisation.

Several small language and grammar errors were noted and corrections suggested below:

Line 218 'executive function such as'

Line 236 'we expected a positive/negative effect on cognitive function'

Line 259 sentence needs to be revised as 'excluded were inpatients with aphasia to an extent that participation in the study is not possible' is incorrect use of tenses and unclear.

Line 314 to 321 - alertness subtest and divided attention tests are not explained clearly. Consider expanding these explanations or clarifying the language as the current wording is not reader friendly for those who are not already aware of these tests.

Line 339 - 'higher scores indicate higher levels of concerns about falling' OR 'higher scores indicate a higher level of concern about falling'

426 'effects on cognition'

428 'further research on this specific area? Field?...'

With these small revisions I believe that the paper will prove an interesting manuscript to readers.

Reviewer #2: This paper reports on the secondary endpoint measures of cognitive function from a randomized trial of conventional physiotherapy (cPT) vs CPT plus Square-Stepping Exercise (SSE) in patients undergoing inpatient early geriatric rehabilitation.

Several aspects of this study limit one's ability to draw meaningful conclusions. In particular:

- The sample size is quite small. It is said to have been derived for testing the primary endpoint (Short Physical Performance Battery), but no details of that calculation or the underlying assumptions are provided. To fully interpret the results in this present manuscript, it would be important to know what a clinically meaningful difference would be for the secondary endpoints reported here and what power this sample size would have yielded to observe those differences. The statistical analysis appears to have been carried out appropriately in a technical sense, but some conclusions may be overstated.

- The short duration of the intervention (3 weeks) and the short training duration (30 minutes per session) may simply be too little to have an impact on cognitive function.

The preliminary disclosure pages provide inconsistent information regarding the availability of the data. The response to the data availability section states "Yes - all data are fully available without restriction" but later in that section, the authors indicate that the data cannot be made publicly available and are available for "researchers who meet the criteria..."

The manuscript needs careful proofreading. For example, lines 230-231 are not a sentence. (It appears that the word 'as' should be deleted.) In line 428 "lager" should be "larger".

**Do you want your identity to be public for this peer review?** For information about this choice, including consent withdrawal, please see our Privacy Policy

Reviewer #1: No

Reviewer #2: No

---

## [Author Response · Author response to Decision Letter 1]

2 Sep 2025

Response to reviewer #1: Thank you for taking the time to review our manuscript. We appreciate your constructive comments, which have helped us to improve the clarity and overall quality of the paper.

1. Overall an interesting manuscript. More major concerns involved the data being collected in 2021-2022 but now being published in 2025 - an explanation for the delay and decision to publish now may be helpful. The 40 references are also all from 2023 or before and the majority were published pre 2020 and therefore pre COVID. It would be helpful to update some of the references to make the manuscript as relevant to the present as possible.

Response: Thank you for your valuable feedback, and the careful reading. We would like to clarify that although data collection for the study was conducted between 2021 and 2022, initial findings were already published in April 2024. This prior publication is referenced in the manuscript, e.g. in line 203.

The current manuscript was first submitted to PLOS in March 2025. Based on our experience, this timeline reflects a typical duration for data analysis, interpretation, and manuscript preparation within a research group.

We have revised the reference list and added the following citations to further support the findings of the manuscript:

• Siqueira VAAA, Shigematsu R, Sebastião E. Stepping towards health: a scoping review of square-stepping exercise protocols and outcomes in older adults. BMC Geriatr. 2024;24; 590. doi: 10.1186/s12877-024-05187-8

• Franco-García JM, Pérez-Gómez J, Castillo-Paredes A, Redondo PC, Rojo-Ramos J, Mayordomo-Pinilla N, Villafaina S, Gómez-Álvaro MC, Melo-Alonso M, Carlos-Vivas J. Effects of Twelve Weeks of Square Stepping Exercises on Physical and Cognitive Function and Plasma Content of SMP30: A Randomised Control Trial. Geriatr. 2025; 10: 22. doi:10.3390/geriatrics10010022

2. The text describes how. 72 patients were included from 307 patients and Figure 1 indicates that 192 were excluded due to not meeting inclusion criteria. However, it would be helpful to understand which criteria the patients failed if this data is available. If nearly 2/3 of patients were excluded for not meeting the inclusion criteria then greater detail is required to ensure that there was not a selection bias hidden within these exclusions. It would also be helpful if the authors provided some examples of the 18 'other reasons' why patients were not included for randomisation.

Response: From the beginning, it was evident that the SSE training program was not applicable to all patients due to their varied diagnoses. The most common reason patients were unable to participate was that they could not walk ten meters independently. This is due to the diverse population of patients in the early stages of geriatric rehabilitation. This population consists of patients with acute neurological and orthopedic conditions. Due to paresis after stroke and fractures with partial weight-bearing, meeting this inclusion criterion was impossible. Because of this heterogeneity, the functional levels of patients in early geriatric rehabilitation vary. The SSE is intended for approximately one-third of patients with a higher functional level. Unfortunately, it is impossible to create a detailed analysis based on the data.

3. Several small language and grammar errors were noted and corrections suggested below:

Line 218 'executive function such as'

Response: The word 'such' was added (L219)

Line 236 'we expected a positive/negative effect on cognitive function'

Response: The word 'positive' was added (L 238)

Line 259 sentence needs to be revised as 'excluded were inpatients with aphasia to an extent that participation in the study is not possible' is incorrect use of tenses and unclear.

Response: The wording has been replaced by 'Inpatients with severe aphasia, …, were excluded' (LL263 – 266).

Line 314 to 321 - alertness subtest and divided attention tests are not explained clearly. Consider expanding these explanations or clarifying the language as the current wording is not reader friendly for those who are not already aware of these tests.

Response: The description has been revised in LL321-331 and made more detailed.

Line 339 - 'higher scores indicate higher levels of concerns about falling' OR 'higher scores indicate a higher level of concern about falling'

Response: The wording has been replaced by 'higher scores indicate a higher level of concern about falling' (L 348).

426 'effects on cognition'

Response: We have changed accordingly to ‘cognition’ (L 430).

428 'further research on this specific area? Field?...'

Response: Thank your pointing out the missing clarity we have now changed to:

‘As the number of studies on the effect of SSE on cognitive function in older people is still limited, further research in this specific research field with a larger study sample and a longer intervention period is required’ (LL 455-458).

Response to reviewer #2: We are grateful for your careful review and thoughtful suggestions. Your feedback has been helpful in refining the manuscript and addressing important points.

Reviewer #2: This paper reports on the secondary endpoint measures of cognitive function from a randomized trial of conventional physiotherapy (cPT) vs CPT plus Square-Stepping Exercise (SSE) in patients undergoing inpatient early geriatric rehabilitation.

Several aspects of this study limit one's ability to draw meaningful conclusions. In particular:

1. The sample size is quite small. It is said to have been derived for testing the primary endpoint (Short Physical Performance Battery), but no details of that calculation or the underlying assumptions are provided. To fully interpret the results in this present manuscript, it would be important to know what a clinically meaningful difference would be for the secondary endpoints reported here and what power this sample size would have yielded to observe those differences. The statistical analysis appears to have been carried out appropriately in a technical sense, but some conclusions may be overstated.

Response: We also discussed this point in detail in our study group. A reference to the detailed calculation of the sample size in the previous publication can be found in LL 468-471. The results were not reported as they were not part of the statistical analysis of this manuscript. However, they are freely accessible in the cited reference [26].

To the best of our knowledge, there is no established value for a clinically meaningful difference, apart from a statistically significant result, for FES-I (e.g. https://doi.org/10.1093/ageing/afad055) and TAP. Since the VAS was used in EuroQol, no clinically meaningful difference can be derived here.

2. The short duration of the intervention (3 weeks) and the short training duration (30 minutes per session) may simply be too little to have an impact on cognitive function.

Response: We also saw this point as a limiting factor, as shown in L 478. To clarify, the following was added: ‘However, in the specific setting of early geriatric rehabilitation, the duration of intervention is limited to three weeks’ (LL483-484).

3. The preliminary disclosure pages provide inconsistent information regarding the availability of the data. The response to the data availability section states "Yes - all data are fully available without restriction" but later in that section, the authors indicate that the data cannot be made publicly available and are available for "researchers who meet the criteria..."

Response: We are very sorry for the confusion. This must be an error in the submission process. As described in the manuscript: The data that support the findings of this study are available on reasonable request from the corresponding author.

Unfortunately, it is not possible to provide all the data in full because this is a clinical sample taken during normal hospital operations. Some of the collected data comes from patients' medical records. Patients did not agree to make their data available in a repository when they signed the consent form to participate in the study.

4. The manuscript needs careful proofreading. For example, lines 230-231 are not a sentence. (It appears that the word 'as' should be deleted.) In line 428 "lager" should be "larger".

Response: We changed this accordingly in L 456. The manuscript has been proofread thoroughly.

---

## [Decision Letter · Decision Letter 1]

24 Sep 2025

Dear Dr. Koschate-Storm,

Thank you for submitting your manuscript to PLOS ONE. After careful consideration, we feel that it has merit but does not fully meet PLOS ONE’s publication criteria as it currently stands. Therefore, we invite you to submit a revised version of the manuscript that addresses the points raised during the review process.

We look forward to receiving your revised manuscript.

Kind regards,

Mario Ulises Pérez-Zepeda, M.D., Ph.D.

Academic Editor

PLOS ONE

Journal Requirements:

Reviewers' comments:

Reviewer's Responses to Questions

**Comments to the Author**

Reviewer #2: (No Response)

2. Is the manuscript technically sound, and do the data support the conclusions?

Reviewer #2: Partly

3. Has the statistical analysis been performed appropriately and rigorously?

Reviewer #2: Yes

4. Have the authors made all data underlying the findings in their manuscript fully available?

Reviewer #2: No

5. Is the manuscript presented in an intelligible fashion and written in standard English?

Reviewer #2: Yes

Reviewer #2: While the statistical analysis is technically correct, I remain concerned that there is too much potential for selection bias in the small subset of patients included in the study, a lack of sufficient information to draw conclusions about these secondary endpoints from the small (and potentially biased) sample, and the limitations of the intervention and training duration. Combined, these factors make it difficult to reliably interpret the material as presented and thus limit the contribution of this manuscript to the field.

**Do you want your identity to be public for this peer review?** For information about this choice, including consent withdrawal, please see our Privacy Policy

Reviewer #2: No

---

## [Author Response · Author response to Decision Letter 2]

29 Oct 2025

Reviewers' comments:

Reviewer's Responses to Questions

Comments to the Author

1. If the authors have adequately addressed your comments raised in a previous round of review and you feel that this manuscript is now acceptable for publication, you may indicate that here to bypass the “Comments to the Author” section, enter your conflict of interest statement in the “Confidential to Editor” section, and submit your "Accept" recommendation.

Reviewer #2: (No Response)

2. Is the manuscript technically sound, and do the data support the conclusions?

Reviewer #2: Partly

3. Has the statistical analysis been performed appropriately and rigorously?

Reviewer #2: Yes

4. Have the authors made all data underlying the findings in their manuscript fully available?

Reviewer #2: No

Answer

Statement on data availability: German data safety law prohibits the general provision of all collected data. Nevertheless, the data can be made available upon reasonable request from the corresponding author. This has already been declared in the manuscript. With due respect we acknowledge that research data should be made available but German data law prohibits an “unrestricted” public availability. In addition, as this was requested in the ethical clearance that only upon request data will be made available. Therefore, for future researchers they can always ask for the data to be incorporated into their research, this will be no problem.

5. Is the manuscript presented in an intelligible fashion and written in standard English?

Reviewer #2: Yes

6. Review Comments to the Author

Reviewer #2: While the statistical analysis is technically correct, I remain concerned that there is too much potential for selection bias in the small subset of patients included in the study, a lack of sufficient information to draw conclusions about these secondary endpoints from the small (and potentially biased) sample, and the limitations of the intervention and training duration. Combined, these factors make it difficult to reliably interpret the material as presented and thus limit the contribution of this manuscript to the field.

Answer to Reviewers Comment :

We fully understand these concerns and agree with some of them.

The sample size is indeed small, so generalization is not possible. Nevertheless, this is one of the first studies in the field of early geriatric rehabilitation. The study's framework conditions are determined by this setting. It is not comparable to a community-dwelling setting. The public health system defines the intervention period, which is always the same. Although it is shorter than that for community-dwelling individuals, it is defined by law. Since the implementation duration is a general law, this study is comparable to others in the field of geriatrics with the same general law in terms of implementation duration (e.g. Austria).

We acknowledge that the study included a small subset of patients. However, with all due respect, we would like to point out that we collected detailed baseline characteristics (e.g., Barthel, SPPB, MMSE) to clarify the functional level of the patients included in the study. Of course, the results cannot be applied to every geriatric ward worldwide because the functional level can vary. Nevertheless, we believe that this basic information on the patients' functional levels can be generalized because we used popular measurements, such as the SPPB, which is used in geriatric research. In order to carry out this type of intervention in this population, a subsample with a higher functional level had to be formed.

Furthermore, it is important to bear in mind that little research has been conducted in this specific area to date. Despite the small sample size, this is still a randomized controlled study.

In summary, we fully understand these concerns. However, the study was conducted within the legal framework for early geriatric rehabilitation in Germany. This means that the results are also transferable to early geriatric rehabilitation programs that are subject to the same framework conditions (e.g., in Germany and Austria). The functional level of the patients in the subsample is made transparent by the baseline characteristics.

To highlight this more clearly, we have added the following to the imitations: "When interpreting the results, it is also important to note that the inclusion criteria resulted in a small subsample of the geriatric population. This gives rise to a potential selection bias. The results are therefore only transferable to a population with a comparable functional level. " in LL 446-449.

7. PLOS authors have the option to publish the peer review history of their article (what does this mean?). If published, this will include your full peer review and any attached files.

Do you want your identity to be public for this peer review? For information about this choice, including consent withdrawal, please see our Privacy Policy.

Reviewer #2: No

---

## [Decision Letter · Decision Letter 2]

27 Nov 2025

Effects of Square-Stepping Exercise on cognitive function in early geriatric rehabilitation. A randomized controlled explorative study.

PONE-D-25-12909R2

Dear Dr. Koschate-Storm,

We’re pleased to inform you that your manuscript has been judged scientifically suitable for publication and will be formally accepted for publication once it meets all outstanding technical requirements.

Kind regards,

Mario Ulises Pérez-Zepeda, M.D., Ph.D.

Academic Editor

PLOS ONE

Additional Editor Comments (optional):

Reviewers' comments:

Reviewer's Responses to Questions

**Comments to the Author**

Reviewer #2: All comments have been addressed

2. Is the manuscript technically sound, and do the data support the conclusions?

Reviewer #2: (No Response)

3. Has the statistical analysis been performed appropriately and rigorously?

Reviewer #2: (No Response)

4. Have the authors made all data underlying the findings in their manuscript fully available?

Reviewer #2: (No Response)

5. Is the manuscript presented in an intelligible fashion and written in standard English?

Reviewer #2: (No Response)

Reviewer #2: (No Response)

**Do you want your identity to be public for this peer review?** For information about this choice, including consent withdrawal, please see our Privacy Policy

Reviewer #2: No

---

## [Editor Report · Acceptance letter]

PONE-D-25-12909R2

PLOS One

Dear Dr. Koschate-Storm,

I'm pleased to inform you that your manuscript has been deemed suitable for publication in PLOS One. Congratulations! Your manuscript is now being handed over to our production team.

Kind regards,

on behalf of

Dr. Mario Ulises Pérez-Zepeda

Academic Editor

PLOS One